

# Development and validation of UV chromatographic method for quantification of copper and copper nanoparticles in different matrices and pharmaceutical products

Mai A. Fadel[1], Dalia M. A. Elmasry[2], Farida H. Mohamed[3], Asmaa M. Badawy[3] and Hanaa A. Elsamadony[4]

[1] Pharmacology and Pyrogen Unit, Department of Chemistry, Toxicology and Feed Deficiency, Animal Health Research Institute (AHRI), Agriculture Research Center (ARC), Giza, Egypt
[2] Nanomaterials Research and Synthesis Unit, Animal Health Research Institute (AHRI), Agriculture Research Center (ARC), Giza, Egypt
[3] Department of Immunology Research, Animal Health Research Institute (AHRI), Agriculture Research Center (ARC), Giza, Egypt
[4] Department of Poultry Disease and Research, Animal Health Research Institute (AHRI), Agriculture Research Center (ARC), Giza, Egypt

Corresponding author
Mai A. Fadel, dr.mai87@yahoo.com

## ABSTRACT

**Background:** The applications of Cu and CuNPs based on the earth-abundant and inexpensive Cu metal have generated a great deal of interest in recent years, including medical applications. A novel, specific, precise, accurate and sensitive reverse-phase high-performance liquid chromatography (RP-HPLC) method with UV detection has been developed and validated to quantify copper (Cu) and copper nanoparticles (CuNPs) in different biological matrices and pharmaceutical products.

**Methods:** The developed method has been validated for linearity, precision, sensitivity, specificity and accuracy. Cu concentration was detected in pharmaceutical products without an extraction process. Moreover, liver, serum and muscle tissues were used as biological matrices. High Cu recovery in biological samples was afforded by using citric acid as a green chelating agent, exact extraction time and pH adjustment. Cu pharmaceutical and biological samples were eluted by acetonitrile: ammonium acetate (50 mM) with 0.5 mg/ml EDTA (30:70 v:v) as an isocratic mobile phase. EDTA reacted with Cu ions forming a Cu-EDTA coloured complex, separated through the C18 column and detected by UV at 310 nm.

**Results:** The developed method was specific with a short retention time of 4.95 min. It achieved high recovery from 100.3% to 109.9% in pharmaceutical samples and 96.8–105.7% in biological samples. The precision RSD percentage was less than two. The method was sensitive by achieving low detection limits (DL) and quantification limits (QL).

**Conclusion:** The validated method was efficient and economical for detecting Cu and CuNPs by readily available chemicals as EDTA and Citric acid with C18 column, which present the best results on RP-HPLC.

## INTRODUCTION

Copper (Cu) is an essential micronutrient due to its vital role in the body's biological and biochemical processes. It is found in all body tissues and plays a role in making red blood cells and maintaining nerve cells and the immune system (*Soetan, Olaiya & Oyewole, 2010*). However, Cu is very toxic in excessive doses and leads to some metabolic disorders and more tissue accumulation and damage (*Chen et al., 2021*; *Ognik et al., 2016*).
In recent years, copper nanoparticles (CuNPs) have had a strong focus on health-related processes since they possess antibacterial properties and antifungal activity besides their catalytic, optical, and electrical properties (*Argueta-Figueroa et al., 2014*). The development of these CuNPs is constantly growing and progressing for future technologies (*Camacho-Flores et al., 2015*).

Recently, some studies declared the effect of nanoparticles in their metallic form as CuNPs to be antiviral (*Tomaszewski, Radomski & Santos-Martinez, 2015*). Nanometric-sized particles are also efficient in drug delivery, ionizing agents and diagnostic imaging. Additionally, a rapid increase in the improper usage of medications such as antibiotics has led the medical field to investigate new alternatives of biocides against infectious diseases (*Patra et al., 2018*). The emergence of the CuNPs analysis lies in the growing area of applications, and it enhances the knowledge of this new material's nature (*Ismail et al., 2019*).

Nano-metals can penetrate biological membranes due to their high physiological solubility and physicochemical properties (*Awaad et al., 2021*). Thus, the data declared by *Escalona et al. (2017)* proved that CuNPs had higher plasma ceruloplasmin activity than cupper sulfide (CuS). In addition, much excretion of CuS than that of CuNPs confirmed that Cu administered as CuNPs was better dissolved than CuS in an acidic environment and probably better absorbed in the digestive tract (*Cholewińska et al., 2018*).

There are several methods for the characterization of CuNPs. One of the standard methods to analyze the shape and size of the CuNPs is the Transmission Electron Microscopy (TEM). Several other methods, such as Dynamic Light Scattering (DLS) and X-Ray Scattering at Small Angles (SAXS), are also used to measure the particle size. Besides this, only the TEM analysis gives authentic images of the morphology and the shape of the nanostructures. The inorganic material's morphological information is collected using an instrument known as the Scanning Electron Microscope (SEM). The most crucial usage of high-resolution EDS/SEM (~100 Å) is to achieve three-dimensional images with large depth fields using a simple sample preparation (*Choudhary et al., 2019*).

Copper colloids and all other metals are usually absorbed in the ultraviolet-visible (UV-Vis) range because of the excitation of surface plasmon resonance (SPR). However, UV-Vis spectroscopy is considered to be a convenient method to characterize CuNPs. On the macroscopic scale, some of the colloidal metal materials are comparably different,

and in the visible region, some give distinct absorption peaks. The metals such as copper, silver, and gold have shown prominent absorption peaks (*Moniri et al., 2017*). Meanwhile, these methods cannot quantify Cu concentration either in its original or nano form in different pharmaceutical products and biological matrices.

Different techniques and instruments accomplished the detection of Cu ions concentration as atomic absorption spectrometry (AAS) (*Wang et al., 2014*), inductively coupled plasma mass spectrometry (ICP-MS) (*Cao et al., 2020*), ion-pair HPLC (*Shen et al., 2006*), and colorimetric methods (*Zeng, Fang & Wang, 2018*; *Xu et al., 2010*; *Ge et al., 2014*). HPLC has high applicability to diverse analytes types, from small organic molecules and ions to large biomolecules and polymers with highly reproducibility, sensitivity, specificity, precision and robustness (*Dong, 2013*). Although UV-Vis detectors are the most common type of detectors used for HPLC because of their relative ease-of-use and high sensitivity, it is not applicable for Cu concentration detection.

This study presents a method a novel insight into the quantification of Cu and CuNPs in biological matrices and pharmaceutical products by the development and validation of the UV-HPLC method.

## MATERIALS AND METHODS

### Detection theory

Cu ions cannot be detected by the UV detector for RP-HPLC by the ordinary extraction method. This study depends on the reaction between Cu ions and EDTA to form a stable complex. EDTA is a strong chelating agent. It can form very stable complexes with the transition elements due to EDTA has hexadentate ligand (*Al-Qahtani, 2017*).

This concept was nearly similar to that in the studies by *Khuhawar & Lanjwani (1995)* and *Rasul Fallahi & Khayatian (2017)*, who used colored reagents to detect metal ions. This Cu-EDTA complex (Fig. 1) is easily detected by the UV detector at 310 nm.

### Standards, drugs, and chemicals

Copper nitrate ($Cu(NO_3)_2$) standard in $HNO_3$ (0.5 mol/l) 1,000 mg/l was purchased from Merck. Acetonitrile (ACN) and methanol (MeOH) were of HPLC grade (Fischer). Ammonium acetate was purchased from Riedel-de Haen (Buchs, SG, Switzerland). Ethylenediaminetetraacetic acid (ETDA) was from Oxford Lab Chem, India. The analytical grade chemicals and reagents were supplied from BDH Laboratories Supplies (BDH Chemical Ltd., Poole, UK). The used pharmaceutical products to check the applicability of the method were Clo-Fix (25 gm/l) by Cardimyer Pharmaceutical Industries (Al Buhayrah, Egypt) and Meracid (10 mg/l) by EVP Co. (Alexandria, Egypt). The CuNPs product was supplied as copper-chitosan nanocomposite (CuCNPs) from the Nanotechnology Research and Synthesis Unit, Animal Health Research Institute (AHRI), Egypt. Copper nanoparticles in extracted matrices were characterized by Transmission electron microscope (TEM) in Central lab., National Research Center (NRC), Egypt. Copper nanoparticles were with an average size 24.71 ± 1.68 nm (Fig. 2).

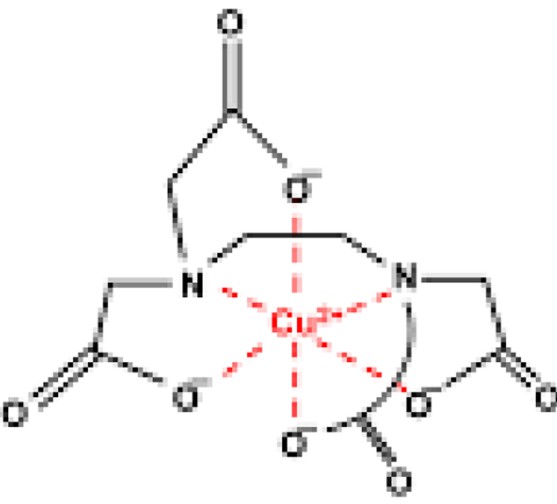

**Cu(II)-EDTA**

**Figure 1  Chemical structure of Cu-EDTA complex.**

## Instrumentation
### HPLC system
Agilent Series 1200 quaternary gradient pump, Series 1200 autosampler, Series 1200 UV-Vis detector, and HPLC 2D ChemStation software (Hewlett-Packard, Les Ulis, France). Agilent C18 column (4.6 mm id, 150 mm, 5 μm particle size).

### Inductively coupled plasma mass spectrometry (ICP-MS)
Thermo ICP–MS model iCAP-RQ.

## Sample preparation
### Pharmaceutical samples
Drug samples were prepared by taking an accurate drug volume in 1% nitric acid to have a final concentration (1 mg/ml). The mobile phase diluted variable concentrations.

### Biological matrices extraction
Liver and muscle (blank) tissues were obtained from SPF chicken aged 28 days. These chickens were supplied from QRD experimental farm for research and scientific service, Giza, Egypt. It is owned by a QVETeh veterinary services company for developing the animal health industry in the Middle East, Cairo, Egypt. The slaughtered chickens were transported in an icebox to be prepared for blank and spiked samples in the laboratory. Tissues were grounded and homogenized and kept at −70 °C until the analysis started. Two grams of tissue samples were weighed. The extraction procedures started with 2 ml 10 mM citric acid with an adjusted pH at 2.3 by 10% sodium hydroxide (NaOH) and

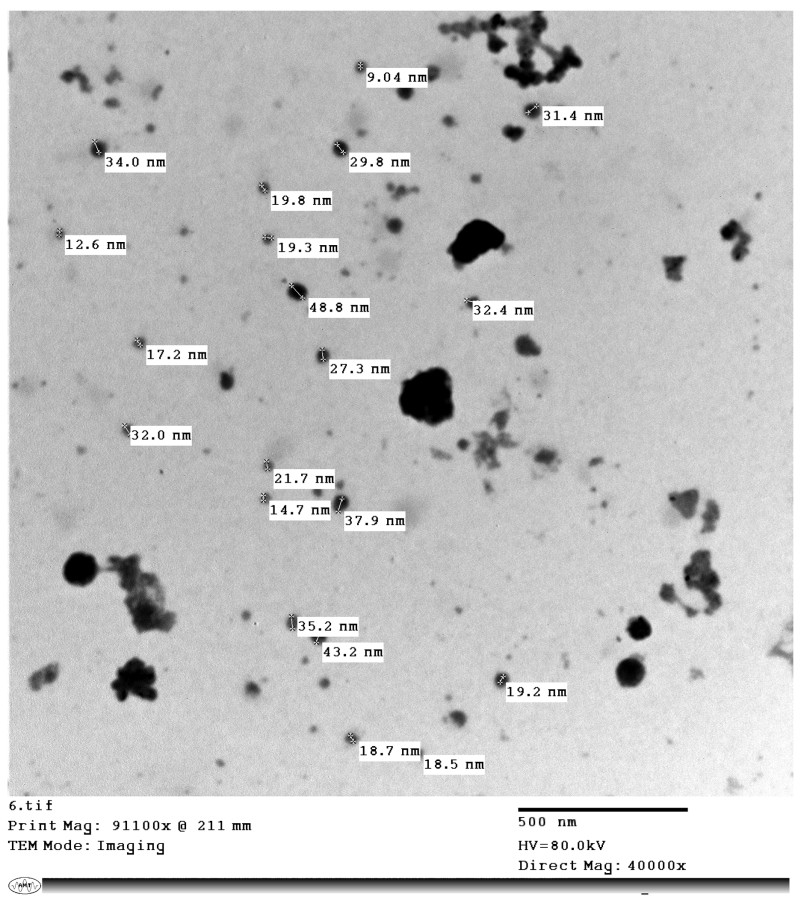

**Figure 2 TEM of Cu-Chitosan nanocomposite showed 24.71 ± 1.68 nm with polydispersity index (PdI): 0.691 ± 0.02, sphere shape and no aggregation.**

2 ml methanolic ammonium acetate (50 mM) with 0.5 mg/ml EDTA. The mixture was vortexed thoroughly for 2 min. They were then shaken for 2 h at 200 rpm at room temperature. Two millilitres of chloroform were also added and mixed well. The samples were centrifuged at 3,000 rpm for 10 min. The organic phase was separated, and another 2 ml of chloroform was added for repetition of the extraction. The extract (4 ml) was evaporated, and the residues were dissolved in the 1 ml mobile phase. The sample (20 µl) was injected onto the HPLC column.

The same procedures were adopted in the serum sample extraction. The difference was in the sample volume (0.5 ml serum) with equal volumes of the other extracted chemicals. Also, the evaporated sample was reconstituted with a 250 µl mobile phase.

## Chromatographic conditions (SIELC Technologies)

The elution mixture consisted of ACN: ammonium acetate (50 mM) with 0.5 mg/ml EDTA (30:70 v:v) as an isocratic mobile phase. The flow rate was 1 ml/min with UV wavelength detection at 310 nm. The stop time was 8 min with a post time of 1 min.

## Standard preparation and calibration curve

The copper stock standard solution was prepared at a 1 mg/ml concentration in 1% nitric acid and diluted to prepare fortified solution at a 100 µg/ml concentration. Calibration standards were prepared at various concentrations (0.05, 0.1, 0.2, 0.4, 0.8, 1, 5, 10, 15, and 20 µg/ml) from the fortified solution by diluting with 1% nitric acid to ascertain the actual concentration of Cu in pharmaceutical products and Cu-nanoparticles preparation (CuNPs).

The calibration curve of biological matrices was prepared by spiking blank samples (serum, liver, and muscle) with various fortified solution concentrations to have calibration samples (0.1, 0.2, 0.4, 0.8, 1, 5, 10, and 15 µg/g).

Quality control samples for pharmaceutical products and nanocomposite were 10 and 1 µg/ml, respectively. Accuracy was determined by the standard addition of 50 ng/ml on three pharmaceutical products levels at 5, 10, and 15 µg/ml and 0.5, 1, and 1.5 µg/ml nanocomposite.

Quality control serum samples were prepared at three different concentrations 0.8, 1.6, and 2.4 µg/ml. Moreover, liver quality control levels were at 1, 2, and 3.5 µg/g and muscle samples at 0.2, 0.4, and 0.6 µg/g.

## Method validation

This was accomplished in concrete terms according to *European Medicines Agency (2005)* and *USP (2019)* as specificity, linearity, and range, precision, recovery, and accuracy, detection limit (DL) and quantification limit (QL), and robustness and system suitability test (SST).

## Statistical evaluation

The obtained results were analyzed using SPSS Inc., version 22.0, Chicago, IL, the USA to calculate the mean, standard deviation (SD) and the relative standard deviation (RSD) (*Morgan et al., 2019*).

# RESULTS

## High-performance liquid chromatograph
### Method validation

**1.** Linearity, range, precision, accuracy, DL, and QL were illustrated in Table 1.

**Linearity and range**: The linearity of Cu and CuNPs was evaluated by the calibration curve on a range of eight concentrations. The correlation coefficient (R) ranged from 0.9995 to 0.9999.

**Precision:** The precision of a method is the degree of agreement among individual test results when the procedure is applied repeatedly to multiple samplings. It is carried out on intra-day and inter-day precisions. It is expressed as the relative standard deviation (coefficient of variance, CV) of a series of measurements. Intra-day precision was performed on six replicates of the analyte on the same day. Inter-day precision was performed on different days and by different analyzers. Intra- and inter-day precisions RSD percentage in dosage form were evaluated and found to be 0.696 and 1.04,

**Table 1 Validation parameters results of Cu analytical method.**

| Parameter | Cu standard | Cu in serum | Cu in liver | Cu in muscle |
|---|---|---|---|---|
| Range (ppm) | 0.05–20 | 0.1–15 | 0.1–15 | 0.1–15 |
| Regression equation | $y = 72.149x - 0.104$ | $y = 73.234x + 0.128$ | $y = 73.263x + 0.3742$ | $y = 75.323x + 0.0594$ |
| Correlation coefficient($r^2$) | 0.9999 | 0.9999 | 0.9995 | 0.9998 |
| Intraday precision (RSD%) | 0.696 | 0.6 | 0.52 | 0.58 |
| Inter-day precision (RSD%) | 1.04 | 1.18 | 0.85 | 1.36 |
| Recovery% | 99.13–101.01 | 96.8–100.7 | 99.98–104.75 | 98.4–105.7 |
| Accuracy | 100.03 ± 0.46 | 99.4 ± 1.4 | 101.5 ± 1.88 | 101.65 ± 2.3 |
| DL (ppm) | 0.038 | 0.06 | 0.055 | 0.023 |
| QL (ppm) | 0.115 | 0.19 | 0.166 | 0.069 |
| Robustness (pooled RSD%) | 2.22 | 1.1 | 2.23 | 2.25 |

respectively. In biological samples, intra-day precision RSD percentage ranged from 0.52% to 0.6%, while the inter-day precision range was from 0.85% to 1.36%.

**Accuracy (standard addition):** The accuracy of an analytical method is the degree of agreement of test results generated by the actual value. The accuracy must be done on three levels: 50%, 100%, and 150%.

**Sensitivity:** The sensitivity was determined by detection limit (DL) and quantification limit (QL). DL is defined as the lowest concentration at which the instrument can detect but not quantify, and the noise to signal ratio for DL should be 1:3. QL is defined as the lowest concentration at which the instrument can detect and quantify. The noise to signal ratio for LOQ should be 1:10 (*Rao, 2018*; *Uddin, Samanidou & Papadoyannis, 2008*).

**2. Robustness:** It was performed by slight changes of the mobile phase composition, the wavelength of UV, and column temperature, which did not lead to any essential changes in the chromatographic system's performance as specificity and system suitability parameters. The method was robust by calculating pooled RSD percentage for all shifts at a certain concentration level (1 µg/ml), as shown in Table 1. The acceptance criterion of the pooled RSD percentage is ≤6%.

**3. Specificity:** The specificity demonstrated with chromatogram through short, specific retention time (4.955 min), as there was no impurities interference between the extracted samples and pure standard (Fig. 3).

### System suitability test

The method showed to be suitably performed under the optimized conditions, and the RSD percentage was found to be less than 1% for system suitability parameters in Table 2.

### Application of pharmaceutical products

The method developed here was applied to various concentrations (5, 10, and 15 µg/ml) of solutions made from pharmaceutical products for determining the content of Cu and CuNPs (0.5, 1, 1.5 µg/ml). The values of the overall drug percentage recoveries and the RSD value of Cu are 100.3–101.1% and 0.01–0.2%, respectively, and were 103.4–109.9%

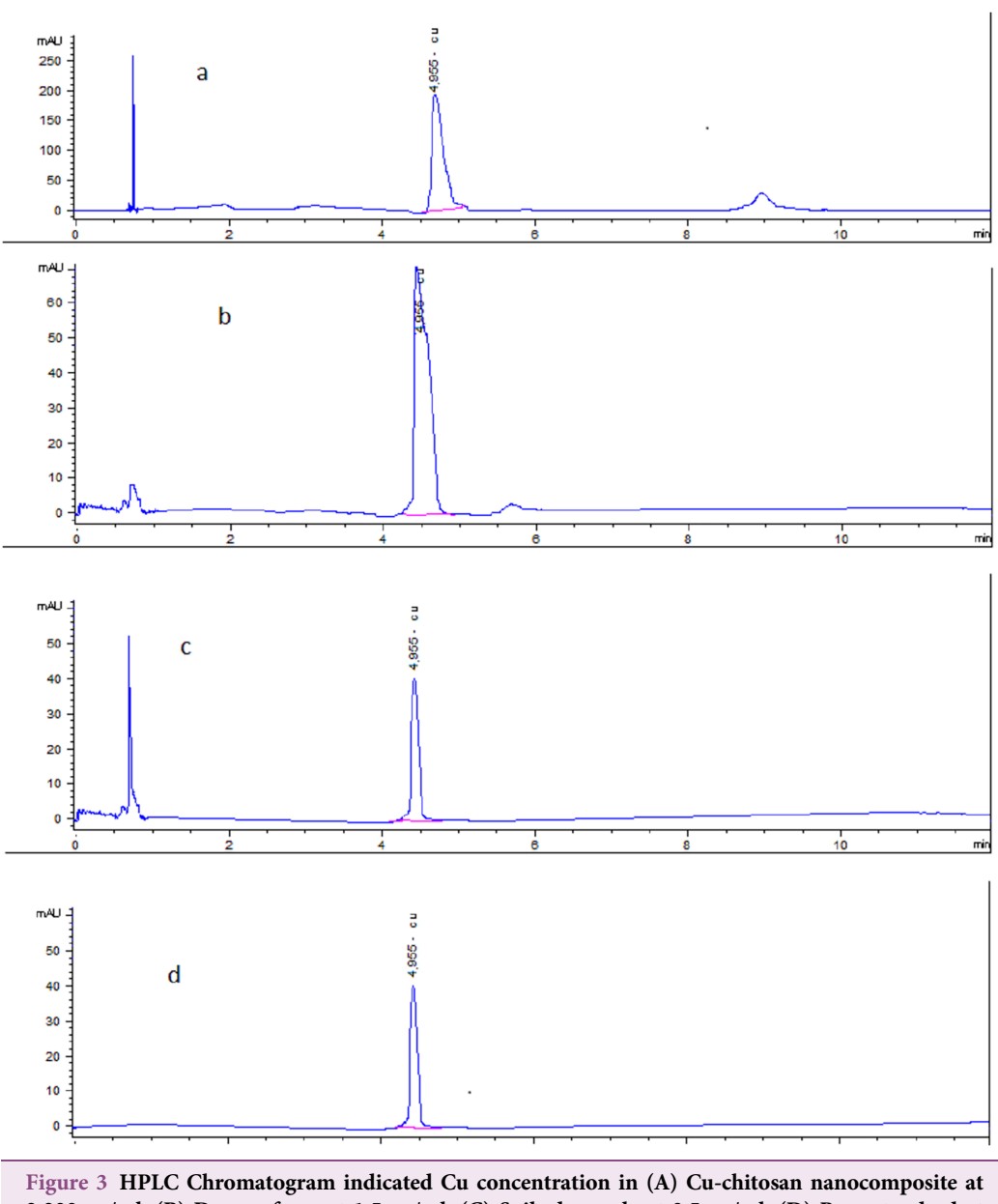

**Figure 3** HPLC Chromatogram indicated Cu concentration in (A) Cu-chitosan nanocomposite at 3.300 μg/ml. (B) Dosage form at 1.5 μg/ml. (C) Spiked muscle at 0.5 μg/ml. (D) Pure standard at 0.5 μg/ml.

**Table 2 System suitability test parameters of Cu at a concentration (1 μg/ml).**

| SST parameters | Cu standard insolvent | Serum | Liver | Muscle | Acceptance criteria |
|---|---|---|---|---|---|
| | Mean ± RSD | | | | |
| Theoretical plates (N) | 4,033.3 ± 0.3 | 4,421 ± 0.3 | 4,231.2 ± 0.106 | 4,264.3 ± 0.92 | >2,000 |
| Retention time (Rt) | 4.95 ± 0.7 | 4.953 ± 0.02 | 4.952 ± 0.005 | 4.955 ± 0.58 | RSD <1.0% |
| Tailing Factor (Tf) | 1.04 ± 0.005 | 1.01 ± 0.1 | 1.024 ± 0.01 | 1.16 ± 0.3 | ≤2.0 |
| Peak area | 72.9 ± 0.81 | 73.6 ± 0.03 | 73.9 ± 0.008 | 75.5 ± 0.45 | RSD <1.0% |
| Symmetry factor | 0.83 ± 0.02 | 0.92 ± 0.02 | 0.71 ± 0.01 | 0.68 ± 0.3 | ≤1.0 |
**Table 3  Pharmaceutical formulations analysis.**

| Analyte | Products name | Added (50 ng/ml)/3 levels | Founded mean ± SD | RSD% | Recovery% | Accuracy ± SD |
|---|---|---|---|---|---|---|
| Copper (Cu) | Clo-Fix (25 gm/l) | 5 µg/ml | 5.045 ± 0.005 | 0.11 | 100.9 | 100.58 ± 0.3 |
| | | 10 µg/ml | 10.06 ± 0.008 | 0.08 | 100.6 | |
| | | 15 µg/ml | 15.038 ± 0.02 | 0.18 | 100.3 | |
| | Meracid (10 mg/l) | 5 µg/ml | 5.05 ± 0.01 | 0.2 | 101.1 | 100.63 ± 0.4 |
| | | 10 µg/ml | 10.05 ± 0.001 | 0.01 | 100.5 | |
| | | 15 µg/ml | 15.05 ± 0.003 | 0.02 | 100.3 | |
| Cu nanoparticle (CuNPs) | CuCNPs preparation (3 g/l) | 0.5 µg/ml | 0.55 ± 0.01 | 1.1 | 109.9 | 106.2 ± 2.9 |
| | | 1 µg/ml | 1.05 ± 0.002 | 0.2 | 105.4 | |
| | | 1.5 µg/ml | 1.55 ± 0.002 | 0.1 | 103.4 | |

**Table 4  Comparison of the results obtained by the developed HPLC method and ICP-MS for the determination of copper in spiked biological samples, pharmaceutical product and CuCNPs composite preparation.**

| Sample | Spiked (PPM) | HPLC method | Recovery (%) | ICP/MS (PPM) |
|---|---|---|---|---|
| Serum | 0.5 | 0.503 ± 0.002 | 100.7 | 0.51 ± 0.001 |
| | 1 | 1.029 ± 0.005 | 102.9 | 1.03 ± 0.004 |
| Liver | 0.5 | 0.498 ± 0.006 | 99.7 | 0.501 ± 0.002 |
| | 1 | 1.026 ± 0.003 | 102.6 | 1.022 ± 0.005 |
| Muscle | 0.5 | 0.502 ± 0.002 | 100.5 | 0.499 ± 0.01 |
| | 1 | 1.017 ± 0.01 | 101.8 | 1.012 ± 0.01 |
| Pharmaceutical product (0.5 ppm) | 0.05 | 0.54 ± 0.03 | 98.2 | 0.55 ± 0.1 |
| | 0.1 | 0.603 ± 0.2 | 100.5 | 0.599 ± 0.01 |
| CuCNPs (0.5 ppm) | 0.05 | 0.55 ± 0.01 | 109.9 | 0.54 ± 0.1 |
| | 0.1 | 0.598 ± 0.02 | 99.7 | 0.601 ± 0.2 |

and 0.1–1.1% for CuNPs as presented in Table 3, indicating that these values are acceptable and the method is accurate and precise. Furthermore, there was no interference and no degradation products. High specificity of this method was confirmed by absence of interference from the sample excipients at the detection wavelength.

### Inductively coupled plasma mass spectrometry

Biological samples and pharmaceutical products were analyzed for copper detection by ICP-MS (Varian 810/820-MS ICP Mass Spectrometer) to ensure results of HPLC. Results illustrated in Table 4.

## DISCUSSION

Copper (Cu) is an essential trace element but causes toxic effects with high doses. In this study, Cu and CuNPs were detected by a precise, accurate, and selective UV RP-HPLC method in dosage form and different matrices (serum, liver, and muscles). Cu detection in pharmaceutical products was done without extraction procedures. It is an economical method as fewer solvents and rapid detection are used. This is unlike the findings by *Takele*

*et al. (2017)*, who detected bromazepam-copper (II) complex through an extraction process. The developed method for Cu detection in biological matrices was validated after some modifications in the original one (*Khuhawar & Lanjwani, 1995*). Copper is one of the heavy metals that requires potent chelating agents like citric acid and EDTA to be extracted and detected (*Jafri, Al-Qahtani & Shay, 2017*). Green chelating agents are readily biodegradable and safer chemicals with less phytotoxicity (*Chauhan, Pant & Nigam, 2015*). In this study, citric acid was used as a green chelating agent for the recovery of copper, which is in accordance with the study by *Asemave (2018)*, who estimated its recovery by 88%. Besides, pH adjustment and time of extraction give higher recovery. Herein, citric acid is a natural organic acid, and the green chelating agents afford safe and powerful extraction and support the concept of sustainable chemistry (*Gómez-Garrido et al., 2018*).

The mixing of the extraction solvent to have EDTA, ammonium acetate, and citric acid gave the best chelating power for the purification of Cu (*Hu et al., 2013*). On keeping this line, EDTA is the most widely used acid in modified forms in extracting cationic micronutrients as $Cu^{1+}$. Copper is an electron donor in its oxidative state ($Cu^{1+}$ and $Cu^{2+}$) in enzyme synthesis and cofactor in ceruloplasmin for iron hemostasis (*Wolf et al., 2015*). The solvent mixture chelated Cu and formed a complex soluble in aqueous-methanolic solution and double extracted by chloroform to be easily transferred from the aqueous phase to the organic one. Chloroform (organic phase) was evaporated to get very stable hexadentate ligand complex, and the mobile phase eluted this complex. The mobile phase had EDTA as the visualizing agent of Cu in the Cu-EDTA coloured complex. This coloured complex is a stable chelate and is detected by UV (*Pati, Zhang & Batley, 2019*). This method is selective to Cu cations rather than the other cations that do not form coloured complexes with EDTA. Cu-EDTA complex was detected through the C18 column, which gave the best results on reverse-phase HPLC.

The developed method is more economical, and there are less health and environmental hazards. This is because of using easily applicable chemicals with RP-HPLC, which is the most conventional chromatography technique. This is in line with the theory of green analytical chemistry, which is part of the concept of sustainable development (*Płotka-Wasylka et al., 2021*). It also minimizes analytical equipment and shortens the time elapsed between conducting analysis and obtaining reliable analyses (*Turner, 2013*).

## CONCLUSION

This advanced RP-HPLC method has been validated for accuracy, precision, linearity, and reproducibility following ICH and USP guidelines. The limits of detection and quantification are very low for both pharmaceutical products and biological matrices. This indicated high sensitivity. RP HPLC is used to quantify copper and Cu NPs *via* potent chelating agents as citric acid and EDTA for its extraction and detection. The Cu-EDTA complex was successfully detected by UV with high selectivity to Cu cations. Cu-EDTA complex was separated by C18 column, which gave the best results on reversed-phase HPLC. The method is novel, simple, sensitive, selective, precise, and accurate analytically for Cu and CuNPs quantification.

## ACKNOWLEDGEMENTS

The authors express sincere gratitude and profound thanks to the staff of the Department of Chemistry at the Animal Health Research Institute for their support and cooperation in completing this study.

### Funding

The authors received no funding for this work.

### Competing Interests

The authors declare that they have no competing interests

### Author Contributions

- Mai A. Fadel conceived and designed the experiments, performed the experiments, analyzed the data, performed the computation work, prepared figures and/or tables, authored or reviewed drafts of the paper, and approved the final draft.
- Dalia M.A. Elmasry performed the experiments, performed the computation work, prepared figures and/or tables, authored or reviewed drafts of the paper, and approved the final draft.
- Farida H. Mohamed performed the experiments, authored or reviewed drafts of the paper, and approved the final draft.
- Asmaa M. Badawy performed the experiments, authored or reviewed drafts of the paper, and approved the final draft.
- Hanaa A. Elsamadony performed the experiments, authored or reviewed drafts of the paper, and approved the final draft.

### Data Availability

The raw measurements are available in the Supplemental File.

### Supplemental Information

Supplemental information for this article can be found online at http://dx.doi.org/10.7717/peerj-achem.14#supplemental-information.

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
