# Peer review of "Development and validation of UV chromatographic method for quantification of copper and copper nanoparticles in different matrices and pharmaceutical products"

_PeerJ Analytical Chemistry, doi:10.7717/peerj-achem.14_

## Round 0.1 · original submission · Major Revisions

The authors need to address the reviewers' comments carefully.

Reviewer 1 ·

Basic reporting

The work concerns the quantification of copper in biological matrices and pharmaceutical preparations. The RP-HPLC method with DAD detection was selected for analysis. The principle of detection is based on the formation of a complex of copper with citric acid and EDTA as known complexing agents. My doubts are raised by the lack of copper specifications, and although the article shows that the authors denote copper in a metallic form and in the form of nanoparticles, it is doubtful, because they use nitric acid for dissolution. No confirmation of the presence of nanoparticles in the tested matrices, e.g. using TEM or SEM. In my opinion, the authors actually mean Cu (II) and therefore in ionic form. The manuscript should be reworded appropriately and accurately name the form they denote. Validation meets the criteria and it is a pity to waste this effort. I don't really understand the choice of method. In fact, simple colorimetry would suffice in this case. What are the benefits of using HPLC which is a separation method as there is only one analyte? Authors should justify their choice.
There is also no comparison of the precision and accuracy of the method with the determinations described so far. Therefore, it is not known what benefits result from the methodology used.
It is difficult to recommend this article for publication in this form.
Fig. 2 - A peak that needs to be identified appears in the chromatogram.
Fig.3- There are actually two unresolved peaks in this chromatogram so the quantification is questionable.
No UV-VIs spectrum of the analyte determined.

Experimental design

Experimental design hould be improved.

Validity of the findings

It should be improved.

Additional comments

see above

Reviewer 2 ·

Basic reporting

English language used is not professional and requires extensive editing with the help of a colleague proficient in English. Some errors are listed below:

Line 209: reference is not appropriate
Abstract:
• This following sentence is not clear to understand: “The use of citric acid as a green chelating agent and extraction time with pH adjustment afforded high Cu recovery in biological samples.”
• Also correct for appropriate symbols in” The precision RSD% was ˂ 2.”

Title:
• Line 2: Use “copper nanoparticles” instead of “nano copper”
Text:
• et al., needs to be in italics in all references
• line 54: ionizing agents and diagnostic imaging
• line 57 and 60: inaccurate sentence structure
• line 84: detectors
• line 86: This study presents a method (this is not a review)
• line 219: Cu1+
• line 239: RP HPLC is used to quantify copper and Cu NPs via potent
• line 211-213: inaccurate sentence structure

Experimental design

A reverse phase HPLC method to quantify copper and copper nanoparticles is reported in this study. However, this is not a novel method. Along with the references provided, SiELC website has a page with similar method using a different column is published on its website : https://www.sielc.com/Application-HPLC-UV-Analysis-of-Copper-Ions-in-Salt-Mixture.html. The authors have used an existing method and validated its use with different biopharmaceutical samples.

Validity of the findings

The validation of the method was executed and data presented accordingly.

Additional comments

Editing is required to accurately describe the results and scope of this study.

---

## Round 0.2 · accepted · Accept

When you review the proofs, please make sure that all the references are formatted consistently.

Reviewer 2 ·

Basic reporting

The manuscript has been edited appropriately to address all the reviewer's comments. The manuscript is now more polished and ready for publication.
Some of the references have the journal name in italics and in other the journal name is not italicized.
Make all the references consistent in formatting.

Experimental design

The experimental design is clear to understand

Validity of the findings

The results are based on the data obtained and discussed accurately.

Additional comments

Publish after editing the references section